# Childhood sexual abuse, alcohol and drug use problems among Black sexual minority men in six US Cities: Findings from the HPTN 061 study

Sylvia Shangani[1]*, Jacob J. van den Berg[2,3], Typhanye V. Dyer[4], Kenneth H. Mayer[5,6], Don Operario[3]

1 Department of Community Health Sciences, School of Public Health, Boston University, Boston, MA, United States of America, 2 Department of Public Health and Community Medicine, School of Medicine, Tufts University, Boston, Massachusetts, United States of America, 3 Department of Behavioral, Social and Health Education, and Social Sciences, Rollins School of Public Health, Emory University, Atlanta, Georgia, United States of America, 4 Department of Epidemiology and Biostatistics, School of Public Health, University of Maryland, College Park, Maryland, United States of America, 5 The Fenway Institute of Fenway Health, Boston, Massachusetts, United States of America, 6 Beth Israel Deaconess Medical Center/Harvard Medical School, Boston, Massachusetts, United States of America

* sylvia65@bu.edu

**Data Availability Statement:** The data cannot be shared publicly without authorization from HPTN HIV Prevention Trial Network. Data are available if

## Abstract

### Background

Prior research has found a high prevalence of childhood sexual abuse (CSA) among sexual minority men (SMM) in the US, and has indicated that CSA is associated with higher rates of alcohol and drug use disorders. However, most of these studies have focused almost exclusively on White SMM. We assessed associations of CSA, alcohol use, and drug use problems among adult Black SMM.

### Methods

Participants were 1,016 Black SMM recruited from six US cities (Atlanta, Boston, Los Angeles, New York City, San Francisco, and Washington, DC) between July 2009 and December 2011. We used hierarchical logistic regression to evaluate the associations between CSA, alcohol use problems ($\geq$ 8 AUDIT), and drug use problems (excluding marijuana).

### Results

Mean (SD) age was 37.8 (11.7) years, and 28.6% and 49.2% reported alcohol and drug use disorders in the past six months, respectively. Most of the study participants reported history of CSA (70.3%). Adjusting for sociodemographic and confounding variables, CSA was associated with higher odds of alcohol use problems (odds ratio (OR) = 1.52, 95% CI 1.09, 2.12) and drug use problems (OR = 1.58, 95% CI 1.19, 2.10) than non-CSA group.

requested at the following web address https://www.hptn.org/research/studies/hptn061/accesstostudydata. There are restrictions to the data and prevent the authors from sharing the de-identified datasets or placing them in a public database. For researchers who meet the criteria for access to confidential data, contact HPTN061 project coordinator at sgiffith@fhi360.org to access the data.

**Funding:** Grant support for HIV Prevention Trials Network (HPTN) 061 grant support was provided by the National Institute of Allergy and Infectious Disease (NIAID), National Institute on Drug Abuse (NIDA) and National Institute of Mental Health (NIMH): Cooperative Agreements UM1 AI068619, UM1 AI068617, and UM1 AI068613. Additional site funding was provided by the Fenway Institute CRS: Harvard University CFAR (P30 AI060354) and CTU for HIV Prevention and Microbicide Research (UM1 AI069480); George Washington University CRS: District of Columbia Developmental CFAR (P30 AI087714); Harlem Prevention Center CRS and NY Blood Center/Union Square CRS: Columbia University CTU (5U01 AI069466) and ARRA funding (3U01 AI069466-03S1); Hope Clinic of the Emory Vaccine Center CRS and The Ponce de Leon Center CRS: Emory University HIV/AIDS CTU (5U01 AI069418), CFAR (P30 AI050409) and CTSA (UL1 RR025008); San Francisco Vaccine and Prevention CRS: ARRA funding (3U01 AI069496-03S1, 3U01 AI069496-03S2); UCLA Vine Street CRS: UCLA, and Department of Medicine, Division of Infectious Diseases CTU (U01 AI069424). The funders had no role in study design, data collection, and preparation of the manuscript.

**Competing interests:** The authors have declared that no competing interests exist.

## Conclusion

Prevalence of CSA is high among BSMM in the US and is positively associated with alcohol and drug use problems. Substance use interventions should address the psychological health needs of BSMM with a history of CSA.

## 1. Introduction

Sexual and gender minorities are at higher risk for problems related to alcohol and drug use compared with heterosexuals [1–3]. Data show that sexual minority adults are more than twice as likely as their heterosexual counterparts to use illicit drugs and almost twice as likely to suffer from a substance abuse disorder [4, 5]. For example, opioid use, including abuse in the past year, was 9% among sexual minority adults compared to 3.8% among the overall adult population [2]. Alcohol and drug use are associated with several adverse health outcomes and comorbidities, including HIV, overdose, mortality, and other sexually transmitted infections [6–8]. Sexual minorities disproportionately suffer from adverse effects of alcohol and drug use and are more likely to have additional comorbidities, including HIV and mental health disorders [9, 10]. BSMM experience more significant risks for these comorbidities, including HIV and STIs, compared with white SMM despite reporting less frequent sexual risk behaviors [11, 12]. The reasons for the disproportionate risks to these conditions among BSMM are due intersecting structural adversities, including social stressors [13]. Understanding factors associated with drug and alcohol use problems among Black SMM is critical, given that this group disproportionately suffer from adverse health effects.

According to the minority stress theory, sexual minority individuals experience multiple stressors, including enacted stigma, prejudice, identity concealment, internalized stigma, and anticipated stigma [13]. As a result of these stressors, some may resort to substance use as a coping mechanism [14]. Prior studies have identified associations between minority stress and increased likelihood of alcohol and drug use among sexual minorities [13, 15, 16]. Also, existing data show that Black SMM are disproportionately affected by minority stressors [6, 17–19]. Black SMM may experience discrimination due to their sexual identity and/or their race/ethnicity. Some studies show that they may use substances such as heroin, cocaine, and methamphetamine to cope with social stress [14, 20]. Black SMM often experience concomitant challenges from poverty, stigma, violence, and high community HIV and STI prevalence that produce a cumulative effect on substance use, sexual risks, and subsequent HIV infection [14, 18]. Several psychosocial stressors are associated with substance use among Black SMM. These include depression, internalized homophobia, and trauma [21–24]. Roots of psychosocial problems among Black SMM have been shown to lie in early experiences of stress, stigma and discrimination, which in turn increase the likelihood of mental health problems, substance abuse, and sexual risk behavior [25].

Childhood sexual abuse (CSA) is among the early childhood stressors reported among Black SMM [18, 26]. CSA has been linked to HIV risk, mental health problems, substance abuse, and intimate partner violence among many populations [27–29]. CSA has garnered attention as a precursor to increased risk for HIV, especially given meta-analytic evidence of elevated rates of CSA among MSM [30]. However, the majority of these studies have focused almost exclusively on White SMM. Limited studies have documented CSA among Black SMM. Notable exceptions include a study of HIV-positive Black SMM in New York City which found CSA to be a precursor to HIV infection, substance misuse, and intimate partner

violence [31]. To build on this literature, we aimed to assess associations of CSA, alcohol use, and drug use problems in a multi-site study of adult Black SMM.

# 2. Materials and methods

## 2.1. Study design

We analyzed baseline data from HPTN 061 study, which has been described elsewhere in detail [32]. Between 2009 and 2011, men from six US cities (Atlanta, Boston, Los Angeles, New York City, San Francisco, and Washington, DC) participated in a feasibility and acceptability study of an intervention to reduce HIV infection among self-identified Black SMM. Men were eligible to participate in the study if they met the following study inclusion criteria: (1) self-identified as a man or were assigned male at birth, and self-identified as Black, African American, Caribbean Black, or multi-ethnic Black, (2) were 18 years of age or older, (3) reported at least one instance of condomless anal intercourse with a man in the past six months, (4) resided in one of the cities where the study was being conducted, (5) did not plan to move away during the study period, and (6) provided written informed consent for the study. Men were ineligible if they were enrolled in any other HIV intervention research study and had been a participant in an HIV vaccine trial. Prescreening to determine eligibility was performed by trained research staff either in person or over the telephone. At the enrollment visit, eligibility was confirmed, and written informed consent was obtained. Participants provided locator information and demographic information to an interviewer and then completed a behavioral assessment using audio computer-assisted self-interview (ACASI) technology. Following the completion of the ACASI assessment, a social and sexual network questionnaire was completed with the interviewer. The institutional review boards (IRB) at all participating institutions approved the HPTN 061 study [32].

## 2.2. Study sample

From a total of 1553 men at baseline, we excluded 438 with missing data on the independent (CSA) and dependent variables (drug and alcohol use). We also excluded 99 participants with missing data on covariates (depression), and this resulted in the analytical sample of 1016 men with complete data (Fig 1).

## 2.3. Measures

**2.3.1. Independent variable.** *2.3.1.1. Childhood sexual abuse (CSA)*. CSA was defined as: (a) having had sexual experiences before the age of 12 years with someone who was five years older, (b) having had unwanted sexual experience between ages 12 and 16 years or had sex between 12 and 16 years with someone who was five years older. Specifically, the following questions were asked of all participants at their baseline enrollment visit: (1) "When you were growing up (before 12 years old), did you experience any sexual experiences? By sexual experience, I mean sexual touching or sexual intercourse."; (2) "Was the person you had the sexual experiences with an adult or someone at least five years older than you?"; (3) "Between the ages of 12 and 16, did you have any unwanted sexual experiences?"; and (4) "Between the ages of 12 and 16, did you have any sexual experiences with an adult or someone who was at least five years older?" These questions were adapted from previous CSA research [26, 33, 34]. Responses were dichotomized: (1) had sex before age 12 years with someone who was at least five years older, and had unwanted sex between ages 12 and 16 years or had sex between 12 and 16 years with someone who was at least five years older, and (2) no CSA.

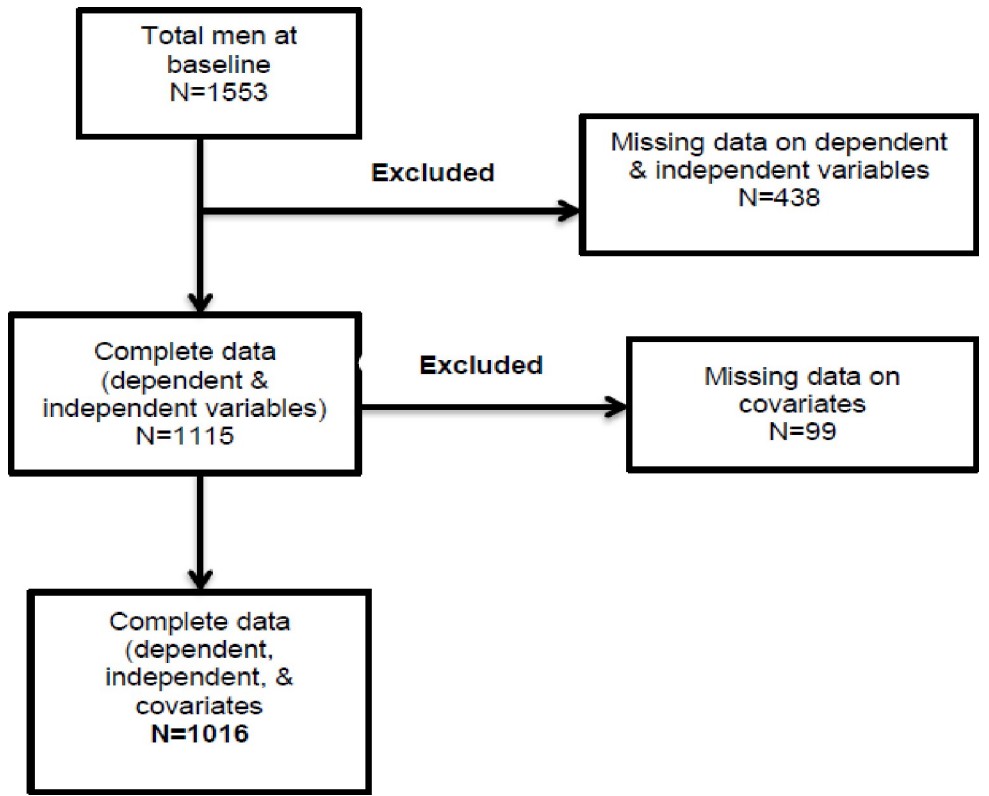

**Fig 1. Analytic sample.**

**2.3.2. Dependent variables.** *2.3.2.1. Drug use problems.* We defined drug use as ever having used any of the following illicit drugs in the past six months: (1) poppers, (2) crack cocaine, (3) powder cocaine, (4) methamphetamine, (5) heroin, (63) unprescribed Vicodin or ketamine, and (7) other non-prescribed drugs. We dichotomized this variable as 0 = "No" and 1 = "Yes" for ever having used any of these substances. Marijuana was excluded as it is pervasive and would not have a rigorous way to characterize abuse.

*2.3.2.2. Alcohol use problems.* We used the Alcohol Use Disorders Identification Test (AUDIT) questionnaire [35], a 10-item questionnaire used to assess alcohol use problems. Items assessed the frequency of drinking and binge drinking, as well as the frequency of alcohol-related problems. Sample items included: "How often do you have a drink containing alcohol?" Four-point Likert scale response options ranged from 0 to 4, with the first response 'never' scoring 0, the second 'less than monthly' scoring 1, the third 'monthly' scoring 2, the fourth 'weekly' scoring 3, and the last response 'daily or almost daily' scoring 4. All scores were then summed, and scores ranged from 0–40. A score of more than eight was used to indicate a problem with alcohol use. We used a dichotomized variable in the analysis (yes = AUDIT score >8) and (no = AUDIT score≤8).

**2.3.3. Covariates.** *2.3.3.1. Depressive symptoms.* We used the Center for Epidemiologic Studies Depression Scale (CES-D) to measure depressive symptoms [36]. Twenty items were asked on a 4-point Likert scale, and a sum was computed for participants who answered all 20 questions on the CES-D. Sample items included: "During the last week, I was bothered by things that usually don't bother me." Four-point response options ranged from "rarely/none of the time (0) to "most of the time (3)". All scores were then summed, and a score of 16 or higher indicated depressive symptoms.

*2.3.3.2. Socio-demographic variables*. Socio-demographic characteristics included age (years), education (less than college, some college or more), employment status (full-time, part-time and not working), student status (full-time student, part-time, and non-student), living situation (respondents were asked how often they experienced insufficient funds to meet essential life needs in the prior six months), income (<$10,000, $10,000–19,999, $20,000–29,000, and >30,000), and relationship status (married or with partner, single, divorced, and separated). Sexual identity was assessed with one item whose 12 response options ("homosexual," "gay," "bisexual," "heterosexual," "same-gender-loving," "sexual," "queer," "two-spirited," "questioning," "polyamorous," "pansexual," and "straight") were collapsed into three categories: homosexual/gay only, bisexual only, and other. Other covariates included sex work (in the last six months), received or provided money or goods for sex), injected drugs in the last six months (yes/no), and have health insurance (yes/no).

## 2.4. Data analysis

This analysis is restricted to participants with complete data on the dependent variables (6-month drug use and alcohol use problems), the independent variable (CSA), and covariates. We used frequencies and percentages to describe the study sample (Table 1). Also, we conducted descriptive statistics by CSA for all the study variables to identify any significant differences between those who reported history of CSA and those with no history of CSA. Chi-square tests were performed for all categorical variables (Table 2). Finally, we conducted hierarchical logistic regression analyses to determine whether CSA was associated with: (1) 6-month drug use and (2) alcohol use problems, adjusting for covariates. We entered CSA status at the first step (model 1), followed by socio-demographic characteristics (model 2), and lastly, depressive symptoms (model 3). The goal was to examine whether the strength of the association between the independent variable (CSA) and the dependent variables (drug use and alcohol use problems) was weakened with further additions of other variables in the model. Statistical analysis was conducted using Stata Statistical Software Release 14 (College Station, TX).

## 3. Results

Table 1 presents the characteristics of the sample ($N$ = 1016). More than half (53.5%) of the study participants were between the ages of 36 and 55 years with a very small percentage of older than 55 years (6%). Almost 50% had ever used at least one of the following drugs: poppers, crack cocaine, powder cocaine, methamphetamine, heroin, unprescribed Vicodin or ketamine, and other non-prescribed drugs in the last six months and close to 30% had a problem with alcohol use in the past six months. Overall, most participants reported experiences of childhood sexual abuse (70%). Also, 43% of the sample reported depressive symptoms.

Drug use during the past six months and alcohol use problems differed by CSA history (refer to Table 2). Participants who had ever experienced any CSA reported higher drug use during the past six months and alcohol use problems compared to those without a history of CSA ($p$<0.001). Participants with a history of CSA were significantly more likely to report depressive symptoms than those without a history of CSA ($p$<0.001). We also observed that participants with a history of CSA were significantly more likely to report insufficient income in the past six months than those with no CSA history *(p = 0.02)*. Additionally, participants with a CSA history were more likely to have provided or received sex for money or drugs in the past six months than those who were not abused *(p<0.001)*.

Tables 3 and 4 show the results of the hierarchical logistic regression analysis. In Table 3, experiences of CSA in model 1 is associated with higher odds of drug use in the past six

**Table 1. Overall characteristics of the study sample (N = 1016).**

| Variable | N (%) or M (SD) |
|---|---|
| **Socio-demographic** | |
| Age (yrs) M(SD) | 37.8 (11.7) |
| Age categories | |
| 18–35 | 414 (40.8) |
| 36–55 | 544 (53.5) |
| >55 | 58 (5.7) |
| Sexual identity | |
| Homosexual/gay only | 352 (34.7) |
| Bisexual only | 313 (30.8) |
| Other | 351 (34.5) |
| Education | |
| Less than college | 550 (54.1) |
| Some college or more | 466 (45.9) |
| Employment status | |
| full-time | 149 (14.7) |
| part-time | 160 (15.7) |
| Not working | 707 (69.6) |
| Income | |
| <10000 | 401 (39.5) |
| 10000–19999 | 221 (21.7) |
| 20000–29000 | 132 (13.0) |
| ≥30000 | 262 (25.8) |
| Not enough money for rent (past six months) | |
| Yes | 568 (55.9) |
| No | 448 (44.1) |
| Relationship status | |
| Married/with partner | 104 (10.2) |
| Single/divorced/separated | 912 (89.8) |
| Student status | |
| Part or full-time student | 200 (19.7) |
| Non-student | 816 (80.3) |
| Health insurance | |
| Yes | 604 (59.4) |
| No | 412 (40.6) |
| **Covariates** | |
| Depressive symptoms | |
| CES-D ≥ (yes) | 439 (43.2) |
| CES-D < 16 (no) | 577 (56.8) |
| Received money/goods for sex | |
| Yes | 236 (23.2) |
| No | 780 (76.8) |
| Provided money/goods for sex | |
| Yes | 102 (10.0) |
| No | 914 (90.0) |
| Injected drugs in the last six months | |
| Yes | 46 (4.5) |
| No | 970 (95.5) |

(*Continued*)

**Table 1.** (Continued)

| Variable | N (%) or M (SD) |
|---|---|
| **Exposure** | |
| CSA | |
| Yes | 713 (70.3) |
| No | 303 (29.8) |
| **Outcome** | |
| Alcohol problem (AUDIT score > 8) | |
| AUDIT score > 8 | 291 (28.6) |
| AUDIT score ≤ 8 | 725 (71.4) |
| Drug use | |
| Yes | 500 (49.2) |
| No | 516 (50.8) |

months compared to the non-CSA group (odds ratio (OR) = 1.71, 95% CI 1.30, 2.24). We included socio-demographic factors (age, sexual identity, education, employment, income, and living conditions with CSA variable in model 2, and found that participants aged 36–55 years and those above 55 years had higher odds of drug use than participants aged 18–35 years, (OR = 2.16, 95% CI 1.64, 2.85) and (OR = 2.34, 95% CI 1.42, 4.16) respectively. Participants who reported insufficient money in the past six months had increased odds of drug use (OR = 1.45, 95% CI 1.11, 1.89). The depressive symptoms variable was added in the final model 3 and was significantly associated with drug use (OR = 1.42, 95% CI = 1.00, 1.70; p = 0.05). The relationship between CSA and drug use remained significant. Also, age and insufficient money for rent in the past six months remained significantly associated with drug use during the past six months.

In Table 4, experiences of CSA in model 1 was associated with 74% increased odds of alcohol use problems. We included socio-demographic factors with CSA variable in model 2. The strength of their association with alcohol use remained significant (p<0.001). In addition, insufficient income in the past six months remained significantly associated with alcohol use problems (OR = 1.66, 95% CI 1.23, 2.24). The depressive symptoms variable was added in the final model 3 and was significantly associated with alcohol use problems (OR = 2.25, 95% CI 1.69, 3.00). The relationship between CSA and alcohol use problem remained significant (OR = 1.52, 95% CI 1.09, 2.12). Participants who reported not enough money for rent in the last six months had 56% increased odds of alcohol use problems (OR = 1.56, 95% CI 1.15, 2.11).

## 4. Discussion

Findings from this six-city study of Black sexual minority men reveal a high prevalence of CSA which was associated with alcohol and drug use problems in the study sample. 70% of the sample had experienced CSA. Overall, 29% and 49% of BSMM reported alcohol use and drug use problems, respectively, and CSA was significantly associated with both risk behaviors independent of sociodemographic and depressive symptoms confounding variables. Our data suggest that interventions to address alcohol and drug use problems among Black SMM should address the role of early life adversities, including the potential role of CSA, as a significant factor contributing to substance use. These findings are particularly relevant in light of national efforts to reduce disparities in substance use among African Americans and to provide them with culturally-relevant intervention and treatment programs [37].

**Table 2. Descriptive summary of study variables by childhood sexual abuse (N = 1016).**

| Variables | CSA (n = 713) N (%) | No CSA (n = 303) M (SD) or N (%) | Statistical test value[a] | P-value |
|---|---|---|---|---|
| **Outcome** | | | | |
| Drug use | | | 14.8 | 0.001 |
| Yes | 379 (53.2) | 121 (39.9) | | |
| No | 334 (46.8) | 182 (60.1) | | |
| Alcohol problem | | | 11.94 | 0.001 |
| AUDIT score > 8 (yes) | 227 (31.8) | 64 (21.1) | | |
| AUDIT score ≤ 8 (no) | 486 (68.2) | 239 (78.9) | | |
| **Socio-demographics** | | | 8.9 | 0.06 |
| Age | | | | |
| 18–35 | 281 (39.4) | 133 (43.9) | | |
| 36–55 | 397 (55.7) | 147 (48.5) | | |
| >55 | 35 (4.9) | 23 (7.6) | | |
| Sexual identity | | | 7.1 | 0.03 |
| Homosexual/gay only | 233 (32.7) | 119 (39.3) | | |
| Bisexual only | 216 (30.3) | 97 (32.0) | | |
| Other | 264 (37.0) | 87 (28.7) | | |
| Education | | | 2.0 | 0.168 |
| Less than college | 396 (55.5) | 154 (50.8) | | |
| Some college or more | 317 (44.5) | 149 (49.2) | | |
| Employment status | | | 4.0 | 0.13 |
| Full-time | 95 (13.3) | 54 (17.8) | | |
| Part-time | 110 (15.4) | 50 (16.5) | | |
| Not working | 508 (71.3) | 199 (65.7) | | |
| Annual Income ($) | | | 9.8 | 0.19 |
| <10000 | 282 (39.5) | 119 (39.3) | | |
| 10000–19999 | 157 (22.0) | 64 (21.1) | | |
| 20000–29000 | 101 (14.2) | 31 (10.2) | | |
| ≥30000 | 173 (24.3) | 89 (29.4) | | |
| Not enough money for rent (past six months) | | | 7.2 | 0.01 |
| Yes | 418 (58.6) | 150 (49.5) | | |
| Never | 295 (41.4) | 153 (50.5) | | |
| Relationship status | | | 0.2 | 0.68 |
| Married/with partner | 75 (10.5) | 29 (9.6) | | |
| Single/divorced/separated | 638 (89.5) | 274 (90.4) | | |
| Student status | | | 0.2 | 0.68 |
| Part or full-time student | 138 (19.3) | 62 (20.5) | | |
| Non-student | 575 (80.7) | 241 (79.5) | | |
| Have health insurance | | | 0.9 | 0.66 |
| Yes | 427 (58.9) | 177 (58.4) | | |
| No | 286 (40.1) | 126 (41.6) | | |
| **Covariates** | | | | |
| Depressive symptoms | | | 14.0 | 0.001 |
| CES-D≥ (yes) | 335 (47.0) | 104 (34.3) | | |
| CES-D<16 (no) | 378 (53.0) | 199 (65.7) | | |
| Received money/goods for sex | | | 7.9 | 0.01 |
| Yes | 183 (25.7) | 53 (17.5) | | |
| No | 530 (74.3) | 250 (82.5) | | |

*(Continued)*

**Table 2.** (Continued)

| Variables | CSA (n = 713) N (%) | No CSA (n = 303) M (SD) or N (%) | Statistical test value[a] | P-value |
|---|---|---|---|---|
| Provided money/goods for sex | | | 3.7 | 0.05 |
| Yes | 80 (22.2) | 22 (7.3) | | |
| No | 633 (88.8) | 281 (92.7) | | |
| Injected drugs in the last six months | | | 1.5 | 0.22 |
| Yes | 36 (5.0) | 10 (3.3) | | |
| No | 677 (95.0) | 293 (96.7) | | |

Note: [a]Statistical test value: Chi Square test, other sexual identity category include: heterosexual same-gender loving, queer, two-spirited, questioning, polyamorous, pansexual, and straight

Our data align with results from a study by Wu (2018) which found 28% CSA prevalence in a New York City sample of 1,002 Black SMM. In that study, CSA history was associated with 50% greater odds of both binge drinking and illicit substance use, which are in line with our findings. However, the prevalence of any CSA was substantially higher in our sample (70%) compared to findings reported in the Wu study, which might reflect a crucial difference in study inclusion criteria. Our study included Black SMM who reported condomless sex during the past six months whereas the study by Wu included Black SMM who reported any sex in the past three months regardless of condom use or nonuse. Indeed, previous research has indicated that CSA is associated with higher rates of condomless sex and HIV risk [29], potentially accounting for the high CSA prevalence in our sample. Moreover, in a previous systematic

**Table 3. Variables associated with drug use: Results from hierarchical logistic regression models (N = 1016).**

| Variables | Model 1 | | Model 2 | | Model 3 | |
|---|---|---|---|---|---|---|
| | OR (95% CI) | P value | OR (95% CI) | P value | OR (95% CI) | P value |
| Exposure (ref-no CSA) | | | | | | |
| CSA | 1.71 (1.30–2.24) | <0.001 | 1.63 (1.22–2.16) | <0.001 | 158 (1.19, 2.10) | 0.002 |
| Socio-demographic factors | | | | | | <0.001 |
| Age (ref-18-35) | | | | | | |
| 36–55 | | | 2.16 (1.64, 2.85) | <0.001 | 2.19 (1.66, 2.89) | <0.001 |
| >55 | | | 2.34 (1.32, 4.16) | <0.001 | 2.36 (1.32, 4.19) | <0.001 |
| Sexual identity (ref-other) | | | | | | |
| Homosexual/gay only | | | 0.88 (0.65, 1.20) | 0.45 | 0.88 (0.64, 1.20) | 0.43 |
| Bisexual only | | | 0.95 (0.69, 1.30) | 0.75 | 0.95 (0.69, 1.31) | 0.77 |
| Education (ref -< college) | | | | | | |
| Some college or more | | | 1.04 (0.79, 1.37) | 0.76 | 1.06 (0.82, 1.40) | 0.64 |
| Employment status (ref-no work) | | | | | | |
| Full-time | | | 0.71 (0.47, 1.08) | 0.11 | 0.72 (0.47, 1.10) | 0.48 |
| Part-time | | | 0.94 (0.65, 1.33) | 0.69 | 0.94 (0.66, 1.38) | 0.78 |
| Annual Income (ref<10000) | | | | | | |
| 10000–19999 | | | 1.09 (0.77, 1.55) | 0.59 | 1.12 (0.78, 1.56) | 0.51 |
| 20000–29000 | | | 0.79 (0.51, 1.20) | 0.23 | 0.82 (0.52, 1.21) | 0.34 |
| ≥30000 | | | 1.01 (0.69, 1.42) | 0.96 | 1.04 (0.70, 1.46) | 0.84 |
| Not enough money for rent (past six months) (ref-never) | | | | | | |
| Yes | | | 1.45 (1.11, 1.89) | 0.01 | 1.42 (1.08, 1.85) | 0.01 |
| Depressive symptoms | | | | | | 0.05 |
| CES-D ≥ 16 (yes) | | | | | 1.30 (1.00, 1.70) | 0.05 |

**Table 4. Variables associated with alcohol use problems: Results from hierarchical logistic regression models (N = 1016).**

| Variables | Model 1 | | Model 2 | | Model 3 | |
|---|---|---|---|---|---|---|
| | OR (95% CI) | P value | OR (95% CI) | P value | OR (95% CI) | P value |
| Exposure (ref-no csa) | | | | | | 0.05 |
| CSA | 1.74 (1.27, 2.40) | <0.001 | 1.66 (1.20, 2.29) | 0.001 | 1.52 (1.09, 2.12) | 0.01 |
| Socio-demographic factors | | | | | | |
| Age (ref-18-35) | | | | | | |
| 36–55 | | | 1.31 (0.96, 1.76) | 0.08 | 1.36 (1.00, 1.85) | 0.05 |
| >55 | | | 0.86 (0.43, 1.67) | 0.64 | 0.81 (0.44, 1.71) | 0.67 |
| Sexual identity (ref-other) | | | | | | |
| Homosexual/gay only | | | 0.79 (0.56, 1.12) | 0.18 | 0.78 (0.55, 1.10) | 0.16 |
| Bisexual only | | | 1.14 (0.81, 1.59) | 0.46 | 1.16 (0.82, 1.63) | 0.39 |
| Education (ref-< college) | | | | | | |
| Some college or more | | | 0.78 (0.58, 1.06) | 0.10 | 0.82 (0.61, 1.12) | 0.21 |
| Employment (ref-no work) | | | | | | |
| Full-time | | | 1.25 (0.79, 1.96) | 0.34 | 1.33 (0.84, 2.11) | 0.22 |
| Part-time | | | 0.92 (0.62, 1.39) | 0.71 | 1.01 (0.66, 1.53) | 0.95 |
| Annual Income (ref<10000) | | | | | | |
| 10000–19999 | | | 0.95 (0.65, 1.39) | 0.80 | 1.00 (0.69, 1.48) | 0.96 |
| 20000–29000 | | | 0.84 (0.53, 1.34) | 0.47 | 0.93 (0.58, 1.48) | 0.76 |
| ≥30000 | | | 1.18 (0.79, 1.75) | 0.42 | 1.27 (0.85, 1.91) | 0.24 |
| Not enough money for rent (past six months) (ref-never) | | | | | | |
| Yes | | | 1.66 (1.23, 2.24) | 0.001 | 1.56 (1.15, 2.11) | 0.004 |
| Depressive symptoms | | | | | | <0.001 |
| CES-D ≥ 16 (yes) | | | | | 2.25 (1.69, 3.00) | <0.001 |

review and meta-analysis of CSA-related risk factors among SMM [30], CSA was associated with an 85% greater odds of recent condomless sex, 54% greater odds of HIV-positive status, as well as being correlated with substance use in seven of eight published studies (meta-analysis of CSA-substance use association was not conducted due to heterogeneity in measurement). Taken together, the current study provides further evidence for a syndemic of CSA, substance use, and HIV risk among Black SMM that warrants consideration for the implementation of interventions to reduce health disparities in this population.

Notably, our findings also bring attention to the role of depression and financial hardship as potential components of the syndemic comprising CSA, substance use, and HIV risk among Black SMM [18]. With regard to our primary research question (on the association between CSA and substance use behaviors), depression and financial hardship variables were considered confounders and thus were adjusted for in regression analyses. However, depression and financial hardship remained significantly associated with CSA in the fully adjusted models, and were each correlated with alcohol use problems and drug use. These persistent associations highlight a need for nuanced, multi-level conceptual models to guide public health interventions that address substance use and HIV risk among Black SMM. In this case, substance use and HIV behavioral risk factors among Black SMM should be understood in the context of economic hardship and psychosocial vulnerability in this group, as well as through the lens of a life course perspective that includes the possibility of CSA or other forms of early-life abuse [14]. Indeed, studies that have integrated the treatment of CSA trauma-related symptoms with HIV risk reduction counseling for SMM have observed HIV risk reduction among SMM [38].

Limitations to this research must be considered. First, as noted, inclusion criteria stipulated that Black SMM participants must have engaged in condomless sex during the past six months, which might have resulted in the high CSA prevalence observed in the sample. Second, self-report data might have been influenced by social desirability bias. In particular, due to associated stigmas and trauma, reports of CSA are especially prone to challenges in disclosure and recall. At the same time, ACASI was used with participants to minimize these effects. Third, the use of cross-sectional data prevents causal inferences, although the timing of the CSA variable temporally precedes all other risk variables reported here. Fifth, the study sites included larger metropolitan areas with high HIV prevalence, thereby limiting generalizability.

In conclusion, these findings underscore the need to consider CSA as a factor in the design and delivery of prevention intervention and treatment programs for alcohol use problems and drug use with Black SMM. Findings from this study suggest a need for more research to more rigorously measure the psychosocial and life course factors that may account for the link between CSA and substance use behaviors among Black SMM, including depression, trauma, as well as economic hardship. Moreover, findings provide further evidence for a syndemic between CSA, substance use, and HIV risk among Black SMM. These findings also suggest need to understand the role of adverse childhood experiences on health outcomes for BSMM. Future studies should examine the role of intersecting social determinants of health and health outcomes of stigmatized and vulnerable populations like BSMM. Also, based on these findings, intersectional inequalities negatively affect health outcomes of stigmatized populations, thus, there is need for interventions that address social and structural factors to reduce health inequities experienced by this population. Also, future interventions should be designed with meaningful input from the BSMM community, to understand the complexity related to these health challenges and to minimize the possibility of stigmatization due to the sensitive nature of CSA.

## Acknowledgments

We thank the HPTN project staff for their support in providing data for this manuscript. We also wish to acknowledge study participants for their participation in HPTN061 project.

## Author Contributions

**Conceptualization:** Sylvia Shangani.

**Data curation:** Sylvia Shangani.

**Formal analysis:** Sylvia Shangani.

**Methodology:** Sylvia Shangani, Jacob J. van den Berg, Typhanye V. Dyer, Don Operario.

**Writing – original draft:** Sylvia Shangani, Jacob J. van den Berg, Typhanye V. Dyer, Kenneth H. Mayer, Don Operario.

**Writing – review & editing:** Sylvia Shangani, Jacob J. van den Berg, Typhanye V. Dyer, Kenneth H. Mayer, Don Operario.

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
