## [Decision Letter · Decision Letter 0]

28 Sep 2022

PONE-D-22-12287

Childhood sexual abuse, alcohol and drug use problems among Black men who have sex with men (MSM) in six US Cities: Findings from the HPTN 061 study

PLOS ONE

Dear Dr. Shangani,

Thank you for submitting your manuscript to PLOS ONE. After careful consideration, we feel that it has merit but does not fully meet PLOS ONE’s publication criteria as it currently stands. Therefore, we invite you to submit a revised version of the manuscript that addresses the points raised during the review process.

Please note that we have only been able to secure a single reviewer to assess your manuscript. We are issuing a decision on your manuscript at this point to prevent further delays in the evaluation of your manuscript. Please be aware that the editor who handles your revised manuscript might find it necessary to invite additional reviewers to assess this work once the revised manuscript is submitted. However, we will aim to proceed on the basis of this single review if possible. 

The reviewer has several requests (see comments below).  Could you please revise the manuscript to carefully address the concerns raised?

We look forward to receiving your revised manuscript.

Kind regards,

Steve Zimmerman, PhD

Associate Editor, PLOS ONE

2.Please update your submission to use the PLOS LaTeX template. The template and more information on our requirements for LaTeX submissions can be found at http://journals.plos.org/plosone/s/latex.

3.We note that the grant information you provided in the ‘Funding Information’ and ‘Financial Disclosure’ sections do not match. When you resubmit, please ensure that you provide the correct grant numbers for the awards you received for your study in the ‘Funding Information’ section.

4.Thank you for stating the following in the Acknowledgments Section of your manuscript:

“HPTN 061 grant support was provided by the National Institute of Allergy and Infectious Disease (NIAID), National Institute on Drug Abuse (NIDA) and National Institute of Mental Health (NIMH): Cooperative Agreements UM1 AI068619, UM1 AI068617, and UM1 AI068613. Additional site funding –Fenway Institute CRS: Harvard University CFAR (P30 AI060354) and CTU for HIV Prevention and Microbicide Research (UM1 AI069480); George Washington University CRS: District of Columbia Developmental CFAR (P30 AI087714); Harlem Prevention Center CRS and NY Blood Center/Union Square CRS: Columbia University CTU (5U01 AI069466) and ARRA funding (3U01 AI069466-03S1); Hope Clinic of the Emory Vaccine Center CRS and The Ponce de Leon Center CRS: Emory University HIV/AIDS CTU (5U01 AI069418), CFAR (P30 AI050409) and CTSA (UL1 RR025008); San Francisco Vaccine and Prevention CRS: ARRA funding (3U01 AI069496-03S1, 3U01 AI069496-03S2); UCLA Vine Street CRS: UCLA Department of Medicine, Division of Infectious Diseases CTU (U01 AI069424).”

“HPTN 061 grant support was provided by the National Institute of Allergy and Infectious Disease (NIAID), National Institute on Drug Abuse (NIDA) and National Institute of Mental Health (NIMH): Cooperative Agreements UM1 AI068619, UM1 AI068617, and UM1 AI068613.  Additional site funding –Fenway Institute CRS: Harvard University CFAR (P30 AI060354) and CTU for HIV Prevention and Microbicide Research (UM1 AI069480); George Washington University CRS: District of Columbia Developmental CFAR (P30 AI087714); Harlem Prevention Center CRS and NY Blood Center/Union Square CRS: Columbia University CTU (5U01 AI069466) and ARRA funding (3U01 AI069466-03S1); Hope Clinic of the Emory Vaccine Center CRS and The Ponce de Leon Center CRS: Emory University HIV/AIDS CTU (5U01 AI069418), CFAR (P30 AI050409) and CTSA (UL1 RR025008); San Francisco Vaccine and Prevention CRS: ARRA funding (3U01 AI069496-03S1, 3U01 AI069496-03S2); UCLA Vine Street CRS: UCLA Department of Medicine, Division of Infectious Diseases CTU (U01 AI069424. The funders had no role in study design, data collection and analysis, decision to publish, or preparation of the manuscript.”

5.In your Data Availability statement, you have not specified where the minimal data set underlying the results described in your manuscript can be found. PLOS defines a study's minimal data set as the underlying data used to reach the conclusions drawn in the manuscript and any additional data required to replicate the reported study findings in their entirety. All PLOS journals require that the minimal data set be made fully available. For more information about our data policy, please see http://journals.plos.org/plosone/s/data-availability.

Reviewers' comments:

Reviewer's Responses to Questions

**Comments to the Author**

1. Is the manuscript technically sound, and do the data support the conclusions?

Reviewer #1: Yes

2. Has the statistical analysis been performed appropriately and rigorously? 

Reviewer #1: Yes

3. Have the authors made all data underlying the findings in their manuscript fully available?

Reviewer #1: Yes

4. Is the manuscript presented in an intelligible fashion and written in standard English?

Reviewer #1: Yes

5. Review Comments to the Author

Reviewer #1: The authors conducted a study on substance use and childhood sexual abuse among Black sexual minority men. Overall the study is interesting and informative. I only have a few recommendations to improve the paper.

1. In general, I recommend using "sexual minority men" (and the abbreviation SMM) rather than "men who have sex with men" throughout. The former is more directly applicable in the context of minority stress theory, as it's more identity centered (and still includes MSM). Note that it's fine to use MSM when referring to a previous study that used that specific term, but otherwise I strongly recommend using SMM.

2. I would explicitly state that the higher risk of alcohol and drug use is related to social and structural stressors in the first sentence, as otherwise it's easy for a reader to skim the first few lines and misunderstand this (even though you otherwise explain this well in paragraph 2).

3. I recommend adding the OR and confidence intervals for income and rent towards the end of the last results paragraph, as it looks like you added them for all the other variables.

4. Minor point: You may want to use BSMM when discussing Black sexual minority men, as the acronym will come up several times throughout the paper.

5. A bit more on policy recommendations, particularly for Black SMM health equity efforts, is recommended.

6. Similarly, a bit more of future research directions is recommended.

7. Minor point: Reference font should be changed to match the main text.

6. PLOS authors have the option to publish the peer review history of their article (what does this mean?). If published, this will include your full peer review and any attached files.

Reviewer #1: No

---

## [Author Response · Author response to Decision Letter 0]

4 Nov 2022

Thank you for this opportunity to resubmit this manuscript to PLOS ONE. Below we respond to each of the comments provided by the reviewer and academic editor. We have bolded all the revisions in the revised manuscript.

Academic editor

We note that the grant information you provided in the ‘Funding Information’ and ‘Financial Disclosure’ sections do not match. When you resubmit, please ensure that you provide the correct grant numbers for the awards you received for your study in the ‘Funding Information’ section

Response: The funding institutes that supported the HIV Prevention Trials Network (HPTN) are National Institute of Allergy and Infectious Diseases (NIAID), National Institutes of Child Health and Human Development (NICH/HD), National Institute of Mental Health (NIMH), Office of AIDS Research, National Institutes of Health (NIH), Department of Health and Human Services under cooperative agreement UM1- AI068619, UM1-AI068617, and UM1- AI068613.

The funders had no role in study design, data collection and analysis, or preparation of the manuscript.

“HPTN 061 grant support was provided by the National Institute of Allergy and Infectious Disease (NIAID), National Institute on Drug Abuse (NIDA) and National Institute of Mental Health (NIMH): Cooperative Agreements UM1 AI068619, UM1 AI068617, and UM1 AI068613. Additional site funding –Fenway Institute CRS: Harvard University CFAR (P30 AI060354) and CTU for HIV Prevention and Microbicide Research (UM1 AI069480); George Washington University CRS: District of Columbia Developmental CFAR (P30 AI087714); Harlem Prevention Center CRS and NY Blood Center/Union Square CRS: Columbia University CTU (5U01 AI069466) and ARRA funding (3U01 AI069466-03S1); Hope Clinic of the Emory Vaccine Center CRS and The Ponce de Leon Center CRS: Emory University HIV/AIDS CTU (5U01 AI069418), CFAR (P30 AI050409) and CTSA (UL1 RR025008); San Francisco Vaccine and Prevention CRS: ARRA funding (3U01 AI069496-03S1, 3U01 AI069496-03S2); UCLA Vine Street CRS: UCLA Department of Medicine, Division of Infectious Diseases CTU (U01 AI069424. The funders had no role in study design, data collection and analysis, decision to publish, or preparation of the manuscript.”

Response: Thank you for this comment. As recommended, we have removed the funding information from the manuscript.

Response: this is the amended statement of funding information “The funding institutes that supported the HIV Prevention Trials Network (HPTN) are National Institute of Allergy and Infectious Diseases (NIAID), National Institutes of Child Health and Human Development (NICH/HD), National Institute of Mental Health (NIMH), Office of AIDS Research, National Institutes of Health (NIH), Department of Health and Human Services under cooperative agreement UM1- AI068619, UM1-AI068617, and UM1- AI068613.

5.In your Data Availability statement, you have not specified where the minimal data set underlying the results described in your manuscript can be found. PLOS defines a study's minimal data set as the underlying data used to reach the conclusions drawn in the manuscript and any additional data required to replicate the reported study findings in their entirety. All PLOS journals require that the minimal data set be made fully available. For more information about our data policy, please see http://journals.plos.org/plosone/s/data-availability.

Response: HPTN have procedures in place, here is the the request

Response: The data that was used for this study are available upon request from the HIV Prevention Trials Network (https://www.hptn.org/research/studies/hptn061/accesstostudydata).There are restrictions to the data and prevent the authors from sharing the de-identified datasets or placing them in a public database.

Response: We have reviewed the references and updated them accordingly

Reviewer

1. In general, I recommend using "sexual minority men" (and the abbreviation SMM) rather than "men who have sex with men" throughout. The former is more directly applicable in the context of minority stress theory, as it's more identity centered (and still includes MSM). Note that it's fine to use MSM when referring to a previous study that used that specific term, but otherwise I strongly recommend using SMM.

Response: As suggested, we have reviewed the manuscript and replaced the term “MSM” with sexual minority men (SMM). 

2. I would explicitly state that the higher risk of alcohol and drug use is related to social and structural stressors in the first sentence, as otherwise it's easy for a reader to skim the first few lines and misunderstand this (even though you otherwise explain this well in paragraph 2).

Response: We agree with the reviewer, we have clarified this in the revised introduction. We have included “The reasons for the disproportionate risks to these conditions among BSMM are due intersecting structural adversities, including social stressors”

3. I recommend adding the OR and confidence intervals for income and rent towards the end of the last results paragraph, as it looks like you added them for all the other variables.

Response: Thank you for pointing this out. We have added odds ratio and confidence intervals for income and rent in the revised results section.

4. Minor point: You may want to use BSMM when discussing Black sexual minority men, as the acronym will come up several times throughout the paper.

Response: Thank you for this suggestion, the revised manuscript to reflect this suggestion.

5. A bit more on policy recommendations, particularly for Black SMM health equity efforts, is recommended.

Response: We appreciate this suggestion. We have included this recommendation in the revised discussion section. The revised section now includes need to focus on interventions that address social and structural factors that exacerbate health inequities in vulnerable and stigmatized populations like BSMM

Similarly, a bit more of future research directions is recommended.

Response: Yes, we have added more details on areas for future research direction in the discussion section. We have added the need for more studies to examine the role of intersecting social determinants of health and health outcomes of vulnerable and stigmatized populations like BSMM.

6. Minor point: Reference font should be changed to match the main text.

Response: Thank you for noting this, we have corrected the reference font to align with the main text.

---

## [Editor Report · Decision Letter 1]

5 Dec 2022

Childhood sexual abuse, alcohol and drug use problems among Black sexual minority men (BSMM) in six US Cities: Findings from the HPTN 061 study

PONE-D-22-12287R1

Dear Dr. Shangani,

We’re pleased to inform you that your manuscript has been judged scientifically suitable for publication and will be formally accepted for publication once it meets all outstanding technical requirements.

Kind regards,

José J. López-Goñi

Academic Editor

PLOS ONE
---

## [Editor Report · Acceptance letter]

13 Dec 2022

PONE-D-22-12287R1 

Childhood sexual abuse, alcohol and drug use problems among Black sexual minority men in six US Cities: Findings from the HPTN 061 study 

Dear Dr. Shangani:

I'm pleased to inform you that your manuscript has been deemed suitable for publication in PLOS ONE. Congratulations! Your manuscript is now with our production department. 

Kind regards, 

on behalf of

Dr. José J. López-Goñi 

Academic Editor

PLOS ONE